# Pre-Therapeutic VEGF Level in Plasma Is a Prognostic Bio-Marker in Head and Neck Squamous Cell Carcinoma (HNSCC)

**DOI:** 10.3390/cancers13153781

**Published:** 2021-07-27

**Authors:** Julia Siemert, Theresa Wald, Marlen Kolb, Isolde Pettinella, Ulrike Böhm, Markus Pirlich, Susanne Wiegand, Andreas Dietz, Gunnar Wichmann

**Affiliations:** Department of Otorhinolaryngology, Head and Neck Surgery, University Hospital Leipzig, Liebigstr, 10-14, 04103 Leipzig, Germany; julia.siemert@medizin.uni-leipzig.de (J.S.); Theresa.Wald@medizin.uni-leipzig.de (T.W.); kolb.marlen@web.de (M.K.); Isolde.pettinella@gmx.de (I.P.); BoehmUlrike@t-online.de (U.B.); Markus.Pirlich@medizin.uni-leipzig.de (M.P.); Susanne.Wiegand@medizin.uni-leipzig.de (S.W.); Andreas.Dietz@medizin.uni-leipzig.de (A.D.)

**Keywords:** head and neck squamous cell carcinoma (HNSCC), head and neck cancer (HNC), prognostic biomarker, vascular endothelial growth factor (VEGF), outcome research, survival, angiogenesis, anti-angiogenesis, biomarker validation, multivariate Cox proportional hazard regression

## Abstract

**Simple Summary:**

In the context of a growing variety in treatment strategies for patients with cancer, especially approaches based on antiangiogenetic pathways, we aimed to identify a useful biomarker for patients with head and neck squamous cell carcinoma (HNSCC). Our experimental results detected vascular endothelial growth factor (VEGF) in patients’ pre-therapeutic plasma, and not serum, which serves as a suitable biomarker for outcome prognostication. Results were validated in an independent cohort, confirming VEGF as an independent predictor (*Pi*) of outcomes in HNSCC patients. Therefore, pre-therapeutic VEGF in plasma may be an attractive biomarker in future HNSCC studies.

**Abstract:**

Vascular endothelial growth factor (VEGF) is centrally involved in cancer angiogenesis. We hypothesized that pre-therapeutic VEGF levels in serum and plasma differ in their potential as biomarkers for outcomes in head and neck squamous cell carcinoma (HNSCC) patients. As prospectively defined in the study protocols of TRANSCAN-DietINT and NICEI-CIH, we measured VEGF in pretreatment serum and plasma of 75 HNSCC test cohort (TC) patients. We analyzed the prognostic value of VEGF concentrations in serum (VEGF_Serum_) and plasma (VEGF_Plasma_) for event-free survival (EFS) utilizing receiver-operating characteristics (ROC). Mean VEGF concentrations in plasma (34.6, 95% CI 26.0–43.3 ng/L) were significantly lower (*p* = 3.35 × 10^−18^) than in serum (214.8, 95% CI 179.6–250.0 ng/L) but, based on ROC (area under the curve, AUC_Plasma_ = 0.707, 95% CI 0.573–0.840; *p* = 0.006 versus AUC_Serum_ = 0.665, 95% CI 0.528–0.801; *p* = 0.030), superiorly correlated with event-free survival (EFS) of TC patients. Youden indices revealed optimum binary classification with VEGF_Plasma_ 26 ng/L and VEGF_Serum_ 264 ng/L. Kaplan–Meier plots demonstrated superiority of VEGF_Plasma_ in discriminating patients regarding outcome. Patients with VEGF_Plasma_ < 26 ng/L had superior nodal (NC), local (LC) and loco-regional control (LRC) leading to significant prolonged progression-free survival (PFS) and EFS. We successfully validated VEGF_Plasma_ according the cut-off <26 ng/L as predictive for superior outcome in an independent validation cohort (iVC) of 104 HNSCC patients from the studies DeLOS-II and LIFE and found better outcomes including prolonged tumor-specific (TSS) and overall survival (OS). Outcomes in TC and iVC combined again was related to VEGF_Plasma_, and multivariate Cox regression revealed that VEGF_Plasma_ was an independent outcome predictor. In HNSCC, pre-therapeutic VEGF_Plasma_ is prognostic for outcomes.

## 1. Introduction

Vascular endothelial growth factor (VEGF), and its main representative isoform VEGFA, is a main driver of angiogenesis, a hallmark of cancer [1,2]. Produced by a multitude of cell types, e.g., megakaryocytes and platelets [3,4,5], neutrophil granulocytes [6,7], T-lymphocytes [8], and also tumor cells [9,10], VEGF interacts with surrounding stroma and directly or indirectly affects cell proliferation and processes for vessel growth [11,12]. In comparison to healthy controls, VEGF concentrations are upregulated in the tissue, serum, and plasma of patients suffering from various cancers [13,14]. Because VEGF is a central player in physiological and pathophysiological vascularization and angiogenesis and is, not only causatively involved, but also reflects oncological processes, it has already been under investigation as a potential biomarker in several cancer entities [15,16,17,18,19].

In line with this, anti-angiogenetic treatment targeting VEGF with monoclonal antibodies, e.g., Bevacizumab, has been successfully established in 1st and 2nd line treatments for numerous cancer entities [20,21]; however, this has not yet been done for head and neck squamous cell carcinoma (HNSCC). There may be various reasons for this and may include prior difficulties to demonstrate a substantial survival benefit justifying acceptance of adverse events [22]. Such difficulties may result from the limited availability of specimen collected alongside clinical trials [23] and, even more relevant, difficulties in comparing the previous literature because of inconsistent standards in VEGF measurements [24]. A clear link between circulating VEGF in HNSCC patient blood and the relevance of pre-therapeutic levels of VEGF for outcomes appears to be underreported. Moreover, biomarkers identifying patients with a higher risk of relapse or being eligible for targeted anti-angiogenetic therapy would be useful and could lead to more individualized and effective treatments for HNSCC patients.

In this study, we established regular sampling of serum and EDTA-anticoagulated plasma from venous blood draws of HNSCC patients participating in either randomized clinical trials or cohort studies with prospective blood sampling. We demonstrate the superiority of circulating VEGF from plasma over serum of therapy-naïve HNSCC patients as a potential biomarker and further validate its potential as an independent predictor (*Pi*) of outcome in an independent cohort.

## 2. Materials and Methods

### 2.1. Patient Description and Samples

Included in this study were two independent cohorts of therapy-naïve patients with histopathologically confirmed HNSCC (ICD-O-M-8070/3, 8071/3), serving as a test (TC) and an independent validation cohort (iVC), from studies approved by the ethics committee of the Medical Faculty of the University Leipzig (Figure 1). All patients provided written informed consent according to the declaration of Helsinki II prior to participation in the trials and cohort studies described below.

TC patients participated in one of two trials, either TRANSCAN-DietINT, a randomized phase II study for tertiary prevention of HNSCC with a dietary intervention (ethic vote 176-15-01062015), or NICEI-CIH, a prospective cohort study to analyze the neoantigen spectrum, immunogenicity, and clinical efficacy of immune-checkpoint inhibitors in HNSCC (ethic vote 341-15-05102015). The iVC patients are a subsample from the LIFE cohort (vote 201-10-12072010) [25,26,27] or DeLOS II trial (vote 166-07-12072006) [28,29].

Serum and plasma samples from venous blood of patients were collected prospectively before treatment at the time of diagnosis, aliquoted and stored at −80 °C until enzyme-linked immunosorbent assay (ELISA) measurements (see below). Biopsies of HNSCC were taken under general anesthesia.

Clinical data including TNM categories [30] and staging according to criteria of Union for International Cancer Control (UICC), patient characteristics, such as their Eastern Cooperative Oncology Group performance scores (ECOG) or Charlson comorbidity scores (CS) [31], as well as their clinical course were taken from the tumor database of the Otorhinolaryngology Department of our university hospital. At date of first contact, we collected data comprising epidemiologic information, including self-reported tobacco smoking, and alcohol consumption. Patient characteristics are shown in Table 1 and Table 2.

### 2.2. Material and Chemicals

Serum-gel and EDTA-plasma S-Monovettes^dfd^ (Sarstedt, Nümbrecht, Germany) were used to collect venous blood samples from patients. Blood samples were centrifuged (2343× *g*, 10 min). Aliquoted serum and plasma were stored at –80 °C until the measuring of VEGF using indirect sandwich-ELISA based on the Human VEGFA ELISA Development Kit (EDK 0709010; 900-K10) from PeproTech GmbH (Hamburg, Germany). Dulbecco’s phosphate buffered saline (PBS) from Biochrom AG (Berlin, Germany) was used for coating the microtiter plates (Greiner Bio-One, Nürtingen, Germany) with 0.5 µg/mL anti-VEGFA antibodies, overnight at 4 °C. PBS containing 0.025% *v*/*v* Tween^®^20 from Sigma-Aldrich (Darmstadt, Germany) was used for washing. After a 60-min blocking step with PBS containing 5% *v*/*v* heat-inactivated fetal calf serum (FCS; Thermo-Fisher Scientific, Waltham, MA, USA), 50 µL of sample (plasma or serum) and a serial dilution of VEGF for calibration were incubated for 120 min, followed by washing and adding biotinylated anti-VEGFA antibodies. After three further washing steps, streptavidin-horseradish conjugate was incubated for 60 min followed by six washing steps before adding tetra-methylbenzidine (TMB) 1-Stepᵀᴹ Ultra (Pierce via Thermo-Fisher Scientific, Waltham, MA, USA).

### 2.3. VEGF Quantification

TMB 1-Stepᵀᴹ Ultra conversion by horseradish peroxidase (HRP) was stopped by adding the same volume of 1 M sulfuric acid. Optical density (OD) was measured at λ_1_ = 450 nm and λ_2_ = 620 nm using a Synergy2 microplate reader equipped with Gen5 software (BioTek Instruments Inc., Winooski, VT, USA). OD (λ_1_–λ_2_) and was converted to concentration (ng/L) according to 4-parameter calibration curves. Lower limit of detection (LLD) and lower limit of quantification (LLQ) were determined in 30 replicates and triplicate log2 dilutions and were <2 ng/L and <4 ng/L, respectively. There were no detectable matrix-related differences in serum versus EDTA-plasma.

### 2.4. Statistical Analysis

SPSS Version 25 (IBM, Corporation, Armonk, New York, NY, USA) was used for statistical analyses. Differences between quantitative parameters were analyzed using *t*-tests and associations between categorical variables examined by Pearson’s chi-square (*Χ*^2^) test. Receiver operating characteristic (ROC) curves served to calculate Youden score and Youden index in order to define the optimum cut-off values for quantitative parameters for binary classification of patients in TC. We measured time-dependent survival parameters from the date of diagnosis to the date of an event. Analyses included overall survival (OS), tumor-specific survival (TSS), event-free survival (EFS), as well as progression-free survival (PFS). OS (the time from diagnosis to death of any cause, censoring patients who remained alive at the end of follow-up), TSS (the time from diagnosis to cancer-related death, censoring patients who remained alive at the end of follow-up or died from other causes), as well as EFS (the time from diagnosis to relapse or death from any cause, censoring patients at the time of the last follow-up who remained alive without signs of any cancer) and PFS (the time from diagnosis to relapse or cancer-related death censoring patients who remained alive at the end of follow-up or who died from other causes) are present censored at 60 months follow-up. Additionally, we assessed the kind of treatment failure to detect a potential link between VEGF and local control (LC), nodal control (NC), loco-regional control (LRC, the combined LC and NC), and distant control (DC).

By censoring all other PFS events, we measured LC as the time of diagnosis to local recurrence or second primary squamous cell carcinoma in the head and neck region. NC was measured from time of diagnosis to relapse in the neck by focusing on squamous cell carcinoma (SCC)-positive lymph node status only. DC was measured from the time of diagnosis to detection of distant metastasis (M1).

Survival parameters in TC, iVC, and the combined cohort (CC) were analyzed using Kaplan–Meier plots for cumulative survival applying log-rank tests. Hazard ratios (HR) were analyzed using multivariate Cox proportional hazard regression models (mCox) applying the conditional logistic regression and bootstrapping utilizing 1000 iterations. We considered 2-sided *p* ≤ 0.05 as significant.

## 3. Results

### 3.1. Patients Characteristics

Table 1 and Table 2 show patients characteristics in detail. In TC, 75 blood samples of 11 female and 64 male patients with HNSCC, more specifically, 9 laryngeal and hypopharyngeal squamous cell carcinoma (LHSCC), 56 oropharyngeal squamous cell carcinoma (OPSCC), 8 oral squamous cell carcinoma (OSCC) and 2 with cancer of another head and neck side, served for measurements of VEGF concentrations in pretherapeutic plasma and serum. Median age was 59.0 (95% CI 58.0–62.6) years. The iVC consisted of 22 female and 82 male HNSCC patients of which 57 patients had LHSCC, 32 OPSCC, and 15 OSCC. Median age was 57.0 (95% CI 56.3–59.9) years and compared well to the TC.

At database lock, median follow-up time in TC and iVC were 26.4 (95% CI 22.8–30.0) and 45.0 (95% CI 39.6–50.4) months (*p* = 9.48 × 10^−6^).

### 3.2. TNM Staging and Outcome in TC

In TC, pre-therapeutic EDTA-plasma and serum samples stored at −80 °C were available from all patients and VEGF measurements were performed using ELISA. VEGF concentrations in plasma (mean VEGF 34.6, 95% CI 26.0–43.3 ng/L) were significantly lower (*p* = 3.35 × 10^−18^) than in serum (214.8, 95% CI 179.6–250.0 ng/L). Based on ROC (area under the curve, AUC_Plasma_ = 0.707, 95% CI 0.573–0.840; *p* = 0.006 versus AUC_Serum_ = 0.665, 95% CI 0.528–0.801; *p* = 0.030) VEGF_Plasma_ was superiorly correlated with EFS of patients and according to the Youden index (the maximum product of sensitivity and specificity observed) achieved 75% sensitivity and 61.8% specificity when 26 ng/L were used as cut-off (Appendix A). Kaplan–Meier plots of cumulative EFS revealed superiority of plasma versus serum VEGF levels (VEGF_Plasma_ 26 ng/L; VEGF_Serum_ 264 ng/L) as a predictive biomarker for outcomes (Table 1). In line with the Youden index for the optimum split of TC patients, Kaplan–Meier analysis (Figure 2) showed that patients with VEGF_Plasma_ > 26 ng/L had significantly worse EFS (*p* = 0.001), PFS (*p* = 0.006), LC (*p* = 0.042), NC (*p* = 0.015), and LRC (*p* = 0.012), and a trend towards impaired DC (*p* = 0.061); however, this was not the case for OS (*p* = 0.744) and TSS (*p* = 0.582). This was not seen after stratification according to VEGF_Serum_ <264 versus >264 ng/L (Appendix A).

### 3.3. Validation of VEGF_Plasma_ Cut-Off for Dicriminating Outcome Groups

Outcome differences in 104 iVC patients (Table 2) confirmed these results (Figure 3), as patients with VEGF_Plasma_ > 26 ng/L demonstrated significantly impaired EFS (*p* = 0.026), PFS (*p* = 0.005), LC (*p* = 0.010), NC (*p* = 0.039), LRC (*p* = 0.007) and even significantly reduced DC (*p* = 0.019), TSS (*p* = 0.003) and OS (*p* = 0.020). Statistical analysis for the CC similarly showed impaired OS (*p* = 0.055), TSS (*p* = 0.046), EFS (*p* = 2.40 × 10^−4)^, PFS (*p* = 1.04 × 10^−4^), LC (*p* = 0.002), NC (*p* = 0.001), LRC (*p* = 3.46 × 10^−4^), and DC (*p* = 0.004) linked to VEGF_Plasma_ > 26 ng/L (Figure 4). Figure 2, Figure 3 and Figure 4 consistently demonstrate an early split of survival curves for patients grouped according to the cut-off value of 26 ng/L VEGF_Plasma_.

### 3.4. VEGF Is an Independent Predictor for Outcome

Using CC data, multivariate Cox proportional hazard regression (mCox) models demonstrate VEGF as a pre-dominant *Pi* for outcome, even after applying bootstrapping. VEGF_Plasma_ > 26 ng/L in this regard is a superior *Pi* compared to well-known classical risk factors for survival of HNSCC patients like tobacco smoking, alcohol consumption, age, and ECOG (Figure 5). VEGF_Plasma_ > 26 ng/L had the highest impact on LC (*HR* 3.12, 95% CI 1.56–6.24; *p* = 1.34 × 10^−3^), DC (*HR* 3.09, 95% CI 1.33–7.19; *p* = 8.87 × 10^−3^), LRC (*HR* 2.97, 95% CI 1.65–5.44; *p* = 3.08 × 10^−4^), NC (*HR* 2.85, 95% CI 1.33–6.10; *p* = 6.90 × 10^−3^), EFS (*HR* 2.46, 95% CI 1.45–4.18; *p* = 8.40 × 10^−4^), and PFS (*HR* 2.40, 95% CI 1.34–4.31; *p* = 3.40 × 10^−3^). However, VEGF_Plasma_ > 26 ng/L was a relevant *Pi* in mCox for TSS and OS, but failed to demonstrate a significant impact on TSS (*HR* 2.22, 95% CI 0.92–5.33; *p* = 0.075) and OS (*HR* 1.75, 95% CI 0.82–3.72; *p* = 0.146) in these models.

## 4. Discussion

According to our results, based on VEGF quantification in plasma and serum of 75 therapy-naïve TC patients, VEGF_Plasma_ is superior to VEGF_Serum_ as a prognostic biomarker for outcomes in EFS (*p* = 0.001), PFS (*p* = 0.006), LC (*p* = 0.042), NC (*p* = 0.015), and LRC (*p* = 0.012). Using the same cut-off for binary classification of 104 iVC patients, we found superior OS (*p* = 0.020), TSS (*p* = 0.003), EFS (*p* = 0.026), PFS (*p* = 0.005), LC (*p* = 0.010), NC (*p* = 0.039), LRC (*p* = 0.007), and DC (*p* = 0.019) for patients with VEGF_Plasma_ < 26 ng/L. The results were confirmed with the 179 CC patients and multivariate Cox regression models.

The VEGF concentrations were measured in plasma and serum obtained during the same blood draw. Both differed substantially, but are comparable to concentrations measured in blood from treatment-naïve HNSCC, reported for either VEGF_Plasma_ [15] or VEGF_Serum_ [32,33]. In our study, no proportionality was detected for VEGF_Serum_ and VEGF_Plasma_ as they were sometimes nearly the same, but there were huge differences between both. Hence, our results would support former assumptions, whereas VEGF_Serum_ may depend on the platelet’s ability to release VEGF during coagulation [34,35,36] and may be modified under anti-coagulant treatment, which is often administered to comorbid HNSCC; VEGF_Plasma_ may reflect the level of circulating VEGF more accurately [7], thereby, more reliably, representing the pre-therapeutic angiogenetic pressure caused by the malignancy.

This would lead to the hypothesis, that, according to our binary classification, a proportion of our HNSCC patients with <264 ng/L VEGF_Serum_ could potentially be wrongly categorized as belonging to the “low VEGF—low risk group” and having superior outcomes, whereas VEGF_Plasma_ > 26 ng/L correctly classifies them as belonging to the “high VEGF—high risk group”. Demonstrating the superiority of VEGF in plasma over serum (Table 1), VEGF_Plasma_ may therefore allow for better outcome prognoses. Otherwise, it points to the role of the VEGF-signaling pathway in HNSCC development and defines VEGF and its receptors as promising targets for targeted therapies.

Angiogenesis is necessary for the shift from tumor dormancy to exponential tumor growth [37] and belongs to the hallmarks of cancer [1,2]. VEGF as potent inducer of angiogenesis leads in combination with basic Fibroblast Growth Factor (FGF) to a higher microvessel density in HNSCC [38]. The immunohistochemically identified VEGF overexpression in tumor tissue is associated with a poorer prognosis (DFS) [39,40] and a higher mortality of HNSCC patients [40,41].

Thereby, VEGF in patient plasma and serum has been under investigation as a potential biomarker for HNSCC [15,32,33,42,43], but decision making regarding sampling plasma or serum for VEGF measurements, with the achievement of biomarker identification, has not been standardized [24] and is inconsistently used, as seen in several studies [15,32]. Nevertheless, elevated levels of VEGF in serum and plasma in comparison to non-cancer control groups have been detected throughout studies [13,14,15] and thereby highlight that VEGF should be considered more for investigations, especially in regards to actual studies and the development of anti-angiogenetic drugs targeting for instance VEGF and its receptors in HNSCC. Indeed, anti-angiogenetic therapies might be based on either targeting VEGF using antibodies, such as bevacizumab, or receptor-tyrosine kinase inhibitors (TKI) of VEGF receptors, for example sorafenib, sunitinib, lenvatinib, or others.

The sole use of bevacizumab or TKI in HNSCC are not yet FDA-approved. As reported by Argiris et al. [22,32], or demonstrated by ASCO 2020 through the presentation of the LEAP-010 study, evaluating the efficacy of lenvantinib in combination with pembrolizumab in patients with HNSCC [44], targeting the VEGF pathway, remains of interest. A combined use with cisplatin-based chemotherapy or immunotherapy may be useful for this cancer entity. In line with this, several clinical trials are currently underway, as reviewed by Micaily et al. [45]. Such trials, and also treatment of HNSCC in the curative setting, may benefit from using VEGF_Plasma_ as a biomarker for treatment stratification.

Biomarker identification often is only a secondary or surrogate endpoint in recently ongoing phase II RCTs. So far, especially in combined use with anti-angiogenetic targeted therapy, VEGF levels have been reported as potential marker in several studies, e.g., for breast cancer [46], in regards of better patients stratification or treatment success. Nevertheless, VEGF in HNSCC patients’ blood has not being approved being a solid biomarker for outcome prognostication yet. This could be linked to differences between VEGF in serum and plasma and the role of anti-coagulation or thrombotic events in modifying VEGF levels observed to be mostly ignored. 

Unfortunately, the very important findings from phase II and III studies from Argiris et al. [22,32] demonstrating the benefit achievable by VEGF targeting were not appropriately perceived by the community of HNSCC specialists. These studies highlighted both, the value of VEGF as biomarker (despite VEGF was measured in serum) as they were able to demonstrate in 1st line treatment of R/M HNSCC a reduction in median VEGF concentrations from baseline 547.7 ng/L to 59.45 ng/L post treatment along with VEGF-targeting by bevacizumab [32] and the improved outcome achieved through VEGF targeting [22]. Dual targeting of EGFR and VEGF pathways, however, offers a potential opportunity for a subgroup of patients unable to receive a cisplatin-based first-line therapy for R/M HNSCC [22,32,42] and represents a potential strategy to improve efficiency, as demonstrated by the studies by Cohen et al. [42] and Argiris et al. [32]. As both studies reported VEGF and VEGFR2 as biomarkers for superior outcomes, using them for treatment stratification could be possible [22,32,42]. Whereas these phase II studies reported lower grade and rate of toxicity compared to EXTREME, the first-line regimen for R/M HNSCC utilizing cisplatin/carboplatin, 5-fluorouracil and cetuximab [47,48], the phase III study by Argiris et al. reported serious (but manageable) side effects occurring in a substantial proportion of patients [22].

Unfortunately, and to the best of our knowledge, no data are available for randomized controlled trial utilizing VEGF targeting in the curative setting and reporting baseline and concentrations of VEGF in context of patient’s clinical outcome. 

Our study has limitations. Despite efforts made to record all data within our prospective studies completely, some patients did not report pack years of tobacco smoking history and daily alcohol consumption, causing missing data regarding these risk factors in TC and iVC. The follow-up in TC was about 3 years and, consequently, few fatal events occurred limiting the chance to make reliable conclusions regarding the impact of circulating VEGF on survival. Hence, all outcome parameters analyzed consistently revealed VEGF_Plasma_ as *Pi*, and we are confident that improved LC, NC, LRC, PFS, and EFS will translate into improved TSS and OS. Compared to other published studies, a strength of our study is the standardization and uniform procedure in blood sampling and storage since 2007, which is necessary for the discovery of plasma biomarkers [49]. The classification characteristics of the cut-off value of 26 ng/L VEGF_Plasma_ obtained based on EFS in the TC were successfully validated for multiple outcome parameters in the iVC. 

Moreover, a further strength of this study were consistent findings of VEGF_Plasma_ being a *Pi* in the CC in all mCox models for outcome parameters and stably remaining a significant *Pi*, even after internal validation with bootstrapping by applying 1000 iterations. The only exception in this regard were OS and TSS based on limited follow-up time in the TC in particular.

In summary, VEGF in plasma is an *Pi* for outcome superior to a variety of well-established outcome predictors, for instance tobacco smoking, p16-positivity and age [50,51,52] based on mCox and applying bootstrapping. Those three covariates were not found to be among the *Pi* in any mCox model whenever VEGF was included in the analyses. Most mCox models identified UICC stages I, II, and III vs. IV as *Pi*, but neither T nor N categories (N category >1 was the only exception in mCox for NC). Furthermore, VEGF_Plasma_ outperformed CS as *Pi* for LRC and LC, as well as alcohol consumption as *Pi* for LRC and TSS (Figure 5).

## 5. Conclusions

In the curative setting, circulating VEGF in plasma of therapy-naïve HNSCC was superior to VEGF in serum as a prognostic biomarker. VEGF levels below 26 ng/L are associated with improved outcome. In the light of clinical trials reporting benefit of a subgroup of R/M HNSCC patients from VEGF-targeting by bevacizumab, targeting the VEGF pathway may have the potential to improve their outcome. Quantification of VEGF in plasma may potentially facilitate identification of patients that are also at risk in the curative setting.

## Figures and Tables

**Figure 1 cancers-13-03781-f001:**
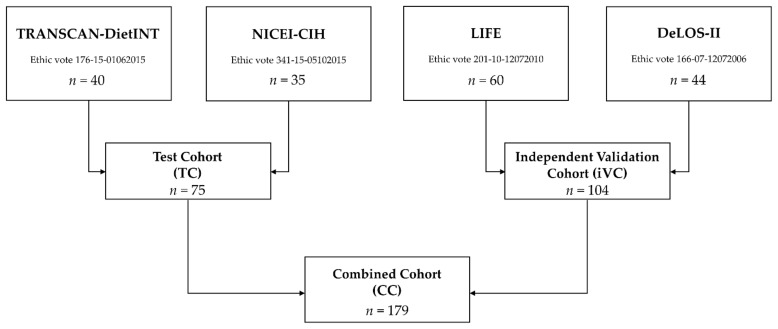
CONSORT diagram illustrating the selection process of *n* = 179 HNSCC patients under study.

**Figure 2 cancers-13-03781-f002:**
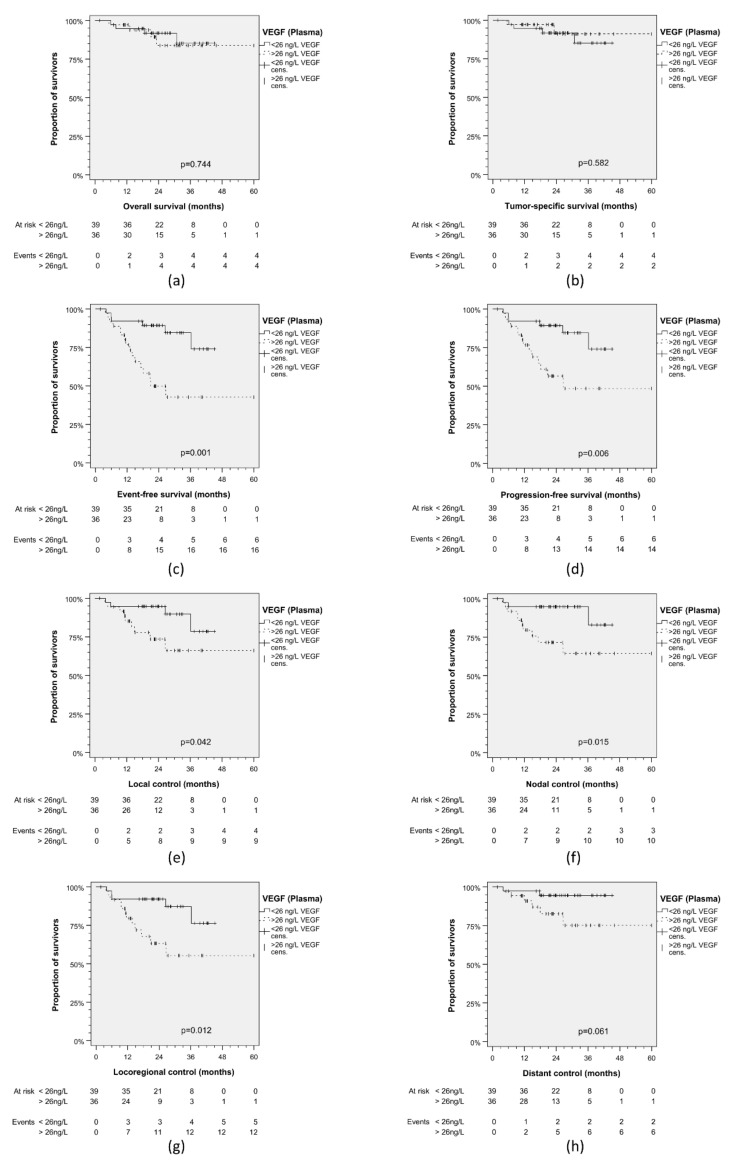
Kaplan–Meier plots for cumulative survival in HNSCC patients from the test cohort stratified according to VEGF in pre-therapeutic plasma. (**a**) Overall survival; (**b**) tumor-specific survival; (**c**) event-free survival; (**d**) progression-free survival; (**e**) local control; (**f**) nodal control; (**g**) loco-regional control; (**h**) distant control. *p* values shown are from 2-sided log-rank test.

**Figure 3 cancers-13-03781-f003:**
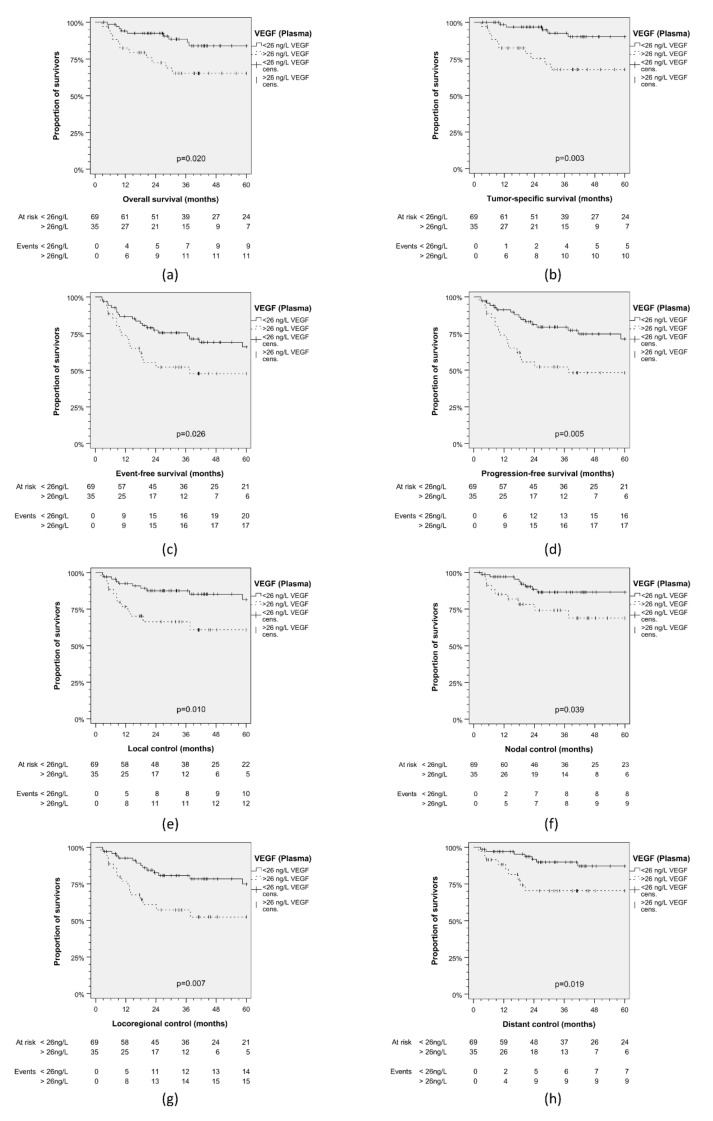
Kaplan–Meier plots for cumulative survival in HNSCC patients from the independent validation cohort stratified according to VEGF in pre-therapeutic plasma. (**a**) Overall survival; (**b**) tumor-specific survival; (**c**) event-free survival; (**d**) progression-free survival; (**e**) local control; (**f**) nodal control; (**g**) loco-regional control; (**h**) distant control. *p* values shown are from 2-sided log-rank test.

**Figure 4 cancers-13-03781-f004:**
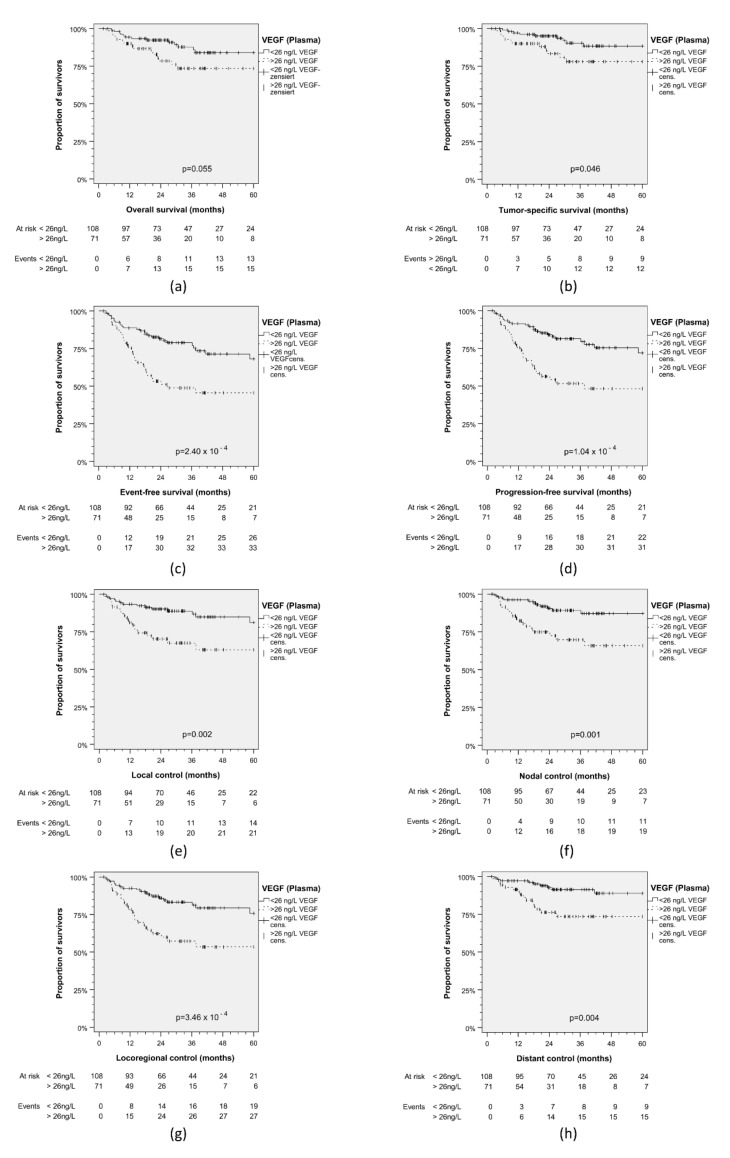
Kaplan–Meier plots for cumulative survival in HNSCC patients from combined test cohort and independent validation cohort stratified according to VEGF in pre-therapeutic plasma. (**a**) Overall survival; (**b**) tumor-specific survival; (**c**) event-free survival; (**d**) progression-free survival; (**e**) local control; (**f**) nodal control; (**g**) loco-regional control; (**h**) distant control. *p* values shown are from 2-sided log-rank test.

**Figure 5 cancers-13-03781-f005:**
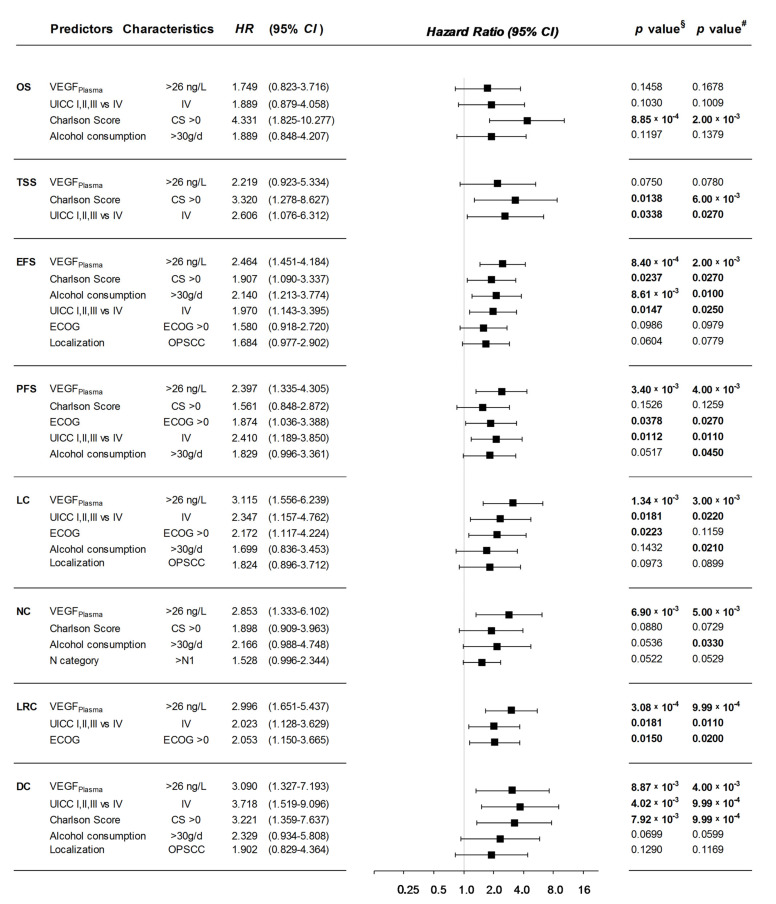
Forrest plots for independent predictors for (OS) overall survival, (TSS) tumor-specific survival, (EFS) event-free survival, (PFS) progression-free survival, (LC) local control, (NC) nodal control, (LRC) loco-regional control, and (DC) distant control from multivariate Cox proportional hazard models build step wise using the likelihood-ratio forward method reaching the highest significance in combined cohorts of HNSCC patients. ^§^ *p* values from multivariate Cox proportional hazard model; *^#^ p* values from multivariate Cox proportional hazard model applying bootstrapping using 1000 iterations.

**Table 1 cancers-13-03781-t001:** Characteristics of HNSCC patients of NICEI-CIH and the Leipzig subsample of TRANSCAN-DietINT constituting the test cohort binary classified based on VEGF in plasma (<26 vs. >26 ng/L) or serum (<264 vs. >264 ng/L). Significant differences between groups (*p* < 0.05) in Pearson’s Chi-square tests are highlighted bold.

Characteristics		TC Cohort		VEGF Plasma					VEGF Serum				
				<26 ng/L		>26 ng/L			<264 ng/L		>264 ng/L		
		*n*	(%)	*n*	(%)	*n*	(%)	*p* Value ^‡^	*n*	(%)	*n*	(%)	*p* Value ^‡^
Study	DietINT	40	(53.3)	24	(61.5)	16	(44.4)	0.1382	29	(56.9)	11	(45.8)	0.3718
	NICEI	35	(46.7)	15	(38.5)	20	(55.6)		22	(43.1)	13	(54.2)	
Sex	female	11	(14.7)	7	(17.9)	4	(11.1)	0.4030	9	(17.6)	2	(8.3)	0.2875
	male	64	(85.3)	32	(82.1)	32	(88.9)		42	(82.4)	22	(91.7)	
BMI ^¶^	15–24.9	36	(48.0)	18	(46.2)	18	(50.0)	0.5578	23	(45.1)	13	(54.2)	0.5227
	25–29.9	32	(42.7)	16	(41.0)	16	(44.4)		22	(43.1)	10	(41.7)	
	>30	7	(9.3)	5	(12.8)	2	(5.6)		6	(11.8)	1	(4.2)	
p16 IHC	negative	15	(20.0)	10	(25.6)	5	(13.9)	0.4453	9	(18.4)	6	(25)	0.5100
(CINtec+)	positive	58	(77.3)	28	(71.8)	30	(83.3)		40	(81.6)	18	(75)	
	unknown	2	(2.7)	1	(2.6)	1	(2.8)		2	(3.9)	--	(--)	
Number of	0	12	(16.0)	7	(17.9)	5	(13.9)	0.7700	8	(15.7)	4	(16.7)	0.6833
positive	1–2	26	(34.7)	14	(35.9)	12	(33.3)		18	(35.3)	8	(33.3)	
nodes	3–4	23	(30.7)	11	(28.2)	12	(33.3)		15	(29.4)	8	(33.3)	
	5–8	10	(13.3)	6	(15.4)	4	(11.1)		6	(11.8)	4	(16.7)	
	>8	4	(5.3)	1	(2.6)	3	(8.3)		4	(7.8)	--	(--)	
Extranodal	negative	22	(29.3)	14	(35.9)	8	(22.2)	0.4714	18	(35.3)	4	(16.7)	0.4399
extension	positive	37	(49.3)	20	(51.3)	17	(47.2)		27	(52.9)	10	(41.7)	
	unknown	16	(21.3)	5	(12.8)	11	(30.6)		6	(11.8)	10	(41.7)	
Grading	1	1	(1.3)	1	(2.6)	--	(--)	0.242	1	(2.0)	--	(--)	0.6376
	2	33	(44.0)	14	(35.9)	19	(52.8)		21	(41.2)	12	(50.0)	
	3	41	(54.7)	24	(61.5)	17	(47.2)		29	(56.9)	12	(50.0)	
Lymphatic	no	7	(9.3)	3	(7.7)	4	(11.1)	0.4659	4	(7.8)	3	(12.5)	0.3472
invasion	yes	54	(72.0)	31	(79.5)	23	(63.9)		40	(78.4)	14	(58.3)	
	unknown	14	(18.7)	5	(12.8)	9	(25.0)		7	(13.7)	7	(29.2)	
Vascular	no	57	(76.0)	31	(79.5)	26	(72.2)	0.6769	41	(80.4)	16	(66.7)	0.2839
invasion	yes	3	(4.0)	2	(5.1)	1	(2.8)		3	(5.9)	--	(--)	
	unknown	15	(20.0)	6	(15.4)	9	(25.0)		7	(13.7)	8	(33.3)	
Perineural	no	51	(68.0)	30	(76.9)	21	(58.3)	0.1564	38	(74.5)	13	(54.2)	0.6237
invasion	yes	9	(12.0)	3	(7.7)	6	(16.7)		6	(11.8)	3	(12.5)	
	unknown	15	(20.0)	6	(15.4)	9	(25.0		7	(13.7)	8	(33.3)	
Any soft risk	no	5	(6.7)	3	(7.7)	2	(5.6)	0.8144	3	(5.9)	2	(8.3)	0.4813
factor	yes	55	(73.3)	30	(76.9)	25	(69.4)		41	(80.4)	14	(58.3)	
	unknown	15	(20.0)	6	(15.4)	9	(25.0)		7	(13.7)	8	(33.3)	
T category	T1	9	(12.0)	6	(15.4)	3	(8.3)	0.4759	7	(13.7)	2	(8.3)	0.1460
TNM 2017	T2	28	(37.3)	15	(38.5)	13	(36.1)		19	(37.3)	9	(37.5)	
	T3	24	(32.0)	13	(33.3)	11	(30.6)		19	(37.3)	5	(20.8)	
	T4	12	(16.0)	4	(10.3)	8	(22.2)		5	(9.8)	7	(29.2)	
	T4a	1	(1.3)	--	(--)	1	(2.8)		--	(--)	1	(4.2)	
	Tx	1	(1.3)	1	(2.6)	--	(--)		1	(2.0)	--	(--)	
N category	0	15	(20.0)	9	(23.1)	6	(16.7)	0.7821	10	(19.6)	5	(20.8)	0.7048
TNM 2017	1	23	(30.7)	13	(33.3)	10	(27.8)		19	(37.3)	4	(16.7)	
	2	21	(28.0)	8	(20.5)	13	(36.1)		12	(23.5)	9	(37.5)	
	2a	3	(4.0)	2	(5.1)	1	(2.8)		2	(3.9)	1	(4.2)	
	2c	3	(4.0)	2	(5.1)	1	(2.8)		2	(3.9)	1	(4.2)	
	3a	3	(4.0)	1	(2.6)	2	(5.6)		2	(3.9)	1	(4.2)	
	3b	7	(9.3)	4	(10.3)	3	(8.3)		4	(7.8)	3	(12.5)	
M category	0	72	(96.0)	38	(97.4)	34	(94.4)	0.5089	50	(98.0)	22	(91.7)	0.1889
TNM 2017	1	3	(4.0)	1	(2.6)	2	(5.6)		1	(2.0)	2	(8.3)	
UICC 2017	I	20	(26.7)	10	(25.6)	10	(27.8)	0.6652	15	(29.4)	5	(20.8)	0.6624
	II	17	(22.7)	11	(28.2)	6	(16.7)		13	(25.5)	4	(16.7)	
	III	22	(29.3)	9	(23.1)	13	(36.1)		14	(27.5)	8	(33.3)	
	IV	3	(4.0)	1	(2.6)	2	(5.6)		1	(2.0)	2	(8.3)	
	IVA	6	(8.0)	4	(10.3)	2	(5.6)		4	(7.8)	2	(8.3)	
	IVB	7	(9.3)	4	(10.3)	3	(8.3)		4	(7.8)	3	(12.5)	
Received	no OP	16	(21.3)	6	(15.4)	10	(27.8)	0.1906	7	(13.7)	9	(37.5)	**0.0191**
surgery	OP	59	(78.7)	33	(84.6)	26	(72.2)		44	(86.3)	15	(62.5)	
Received	no RT	7	(9.3)	2	(5.1)	5	(13.9)	0.1926	4	(7.8)	3	(12.5)	0.5178
radiotherapy	RT	68	(90.7)	37	(94.9)	31	(86.1)		47	(92.2)	21	(87.5)	
Received	no RChT	29	(38.7)	15	(38.5)	14	(38.9)	0.9697	22	(43.1)	7	(29.2)	0.2465
RChT	RChT	46	(61.3)	24	(61.5)	22	(61.1)		29	(56.9)	17	(70.8)	
Event-free	no event	53	(70.7)	33	(84.6)	20	(55.6)	0.0048	38	(74.5)	15	(62.5)	0.2866
survival	event	22	(29.3)	6	(15.4)	16	(44.4)		13	(25.5)	9	(37.5)	
Progression-	no event	55	(73.3)	33	(84.6)	22	(61.1)	0.0215	38	(74.5)	17	(70.8)	0.7370
free survival	event	20	(26.7)	6	(15.4)	14	(38.9)		13	(25.5)	7	(29.2)	
Local control	no event	62	(82.7)	35	(89.7)	27	(75.0)	0.0920	43	(84.3)	19	(79.2)	0.5828
	event	13	(17.3)	4	(10.3)	9	(25.0)		8	(15.7)	5	(20.8)	
Nodal	no event	62	(82.7)	36	(92.3)	26	(72.2)	0.0217	43	(84.3)	19	(79.2)	0.5828
control	event	13	(17.3)	3	(7.7)	10	(27.8)		8	(15.7)	5	(20.8)	
Distant	no event	67	(89.3)	37	(94.9)	30	(83.3)	0.1058	45	(88.2)	22	(91.7)	0.6534
control	event	8	(10.7)	2	(5.1)	6	(16.7)		6	(11.8)	2	(8.3)	
Loco-regional	no event	58	(77.3)	34	(87.2)	24	(66.7)	0.0340	41	(80.4)	17	(67.8)	0.3564
control	event	17	(22.7)	5	(12.8)	12	(33.1)		10	(19.6)	7	(29.2)	
Overall	alive	67	(89.3)	35	(89.7)	32	(88.9)	0.9046	47	(92.2)	20	(83.3)	0.2482
survival	dead	8	(10.7)	4	(10.3)	4	(11.1)		4	(7.8)	4	(16.7)	

^‡^ *p* value from Pearson’s Chi square (χ^2^) tests. ^¶^ Classification of anthropometric height/weight characteristics according to WHO (2000) “Obesity: Preventing and Managing the Global Epidemic” into underweight (BMI 15–19.9 kg/m^2^) and normal weight (BMI 20–24.9 kg/m^2^); overweight (BMI 25–29.9 kg/m^2^); obesity summarizing adiposity I (BMI 30–34.9 kg/m^2^), adiposity II (BMI 35–39.9 kg/m^2^), and adiposity III (BMI ≥ 40 kg/m^2^). p16 IHC (CINtec+), combined immunohistochemical tests for p16^INK4A^ and Ki-67 expression; T, Tumor size; N, Nodal involvement; M, distant metastasis; UICC, Union for International Cancer Control; OP, surgical operation of the primary and/or neck dissection; RT, radiotherapy; RChT, radio-chemo-therapy.

**Table 2 cancers-13-03781-t002:** Characteristics of HNSCC patients in the test cohort (TC; *n =* 75) and independent validation cohort (iVC; *n* = 104). Significant differences between groups (*p* < 0.05) in Pearson’s Chi-square tests are highlighted bold.

Characteristics		All Patients		TC		iVC		
		*n*	(%)	*n*	(%)	*n*	(%)	*p* Value ^‡^
Sex	female	33	(18.4)	11	(14.7)	22	(21.2)	0.2695
	male	146	(81.6)	64	(85.3)	82	(78.8)	
Age Score	<50 years	24	(13.4)	8	(10.7)	16	(15.4)	0.4982
	<60 years	77	(43.0)	31	(41.3)	46	(44.2)	
	<70 years	50	(27.9)	21	(28.0)	29	(27.9)	
	≥70 years	28	(15.6)	15	(20.0)	13	(12.5)	
ECOG	0	118	(65.9)	53	(70.7)	65	(62.5)	0.2554
	>0	61	(34.1)	22	(29.3)	39	(37.5)	
Charlson score	0	98	(54.7)	43	(57.3)	55	(52.9)	0.0722
	>0	81	(45.3)	32	(42.7)	49	(47.1)	
Pack years	≤30 PY	103	(55.3)	44	(58.7)	59	(56.7)	0.0752
	>30 PY	76	(42.5)	31	(41.3)	45	(43.3)	
Alcohol	none	20	(11.2)	7	(9.3)	13	(12.5)	**7.16 × 10^−5^**
consumption	<30 g/d	74	(41.3)	40	(53.3)	34	(32.7)	
	<60 g/d	27	(15.1)	6	(8.0)	21	(20.2)	
	>60 g/d	50	(27.9)	14	(18.7)	36	(34.6)	
	unknown	8	(4.5)	8	(10.7)	--	(--)	
Localization	LHSCC	66	(36.9)	9	(12.0)	57	(54.8)	**4.03 × 10^−9^**
	OPSCC	88	(49.2)	56	(74.7)	32	(30.8)	
	OSCC	23	(12.8)	8	(10.7)	15	(14.4)	
	other	2	(1.1)	2	(2.7)	--	(--)	
OPSCC vs. Other	p16+ OPSCC	70	(39.1)	49	(65.3)	21	(20.2)	**1.02 × 10^−9^**
	other	109	(60.9)	26	(34.7)	83	(79.8)	
T category TNM	T1	26	(14.5)	9	(12.0)	17	(16.3)	**1.06 × 10^−4^**
2017	T2	55	(30.7)	28	(37.3)	27	(26.0)	
	T3	49	(27.4)	24	(32.0)	25	(24.0)	
	T4	18	(10.1)	12	(16.0)	6	(5.8)	
	T4a	29	(16.2)	1	(1.3)	28	(26.9)	
	Tx	2	(0.6)	1	(1.3)	1	(1.0)	
N category TNM	N0	44	(24.6)	15	(20.0)	29	(27.9)	**5.10 × 10^−6^**
2017	N1	42	(23.5)	23	(30.7)	19	(18.3)	
	N2	30	(16.8)	21	(28.0)	9	(8.7)	
	N2a	3	(1.7)	3	(4.0)	--	(--)	
	N2b	14	(7.8)	--	(--)	14	(13.5)	
	N2c	17	(9.5)	3	(4.0)	14	(13.5)	
	N3	3	(1.7)	3	(4.0)	--	(--)	
	N3a	1	(0.6)	--	(--)	1	(1.0)	
	N3b	25	(14.0)	7	(9.3)	18	(17.3)	
M category TNM	M0	175	(97.8)	72	(96.0)	103	(99.0)	0.1748
2017	M1	4	(2.2)	3	(4.0)	1	(1.0)	
UICC 2017	I	38	(21.2)	20	(26.7)	18	(17.3)	**4.73 × 10^−4^**
	II	29	(16.2)	17	(22.7)	12	(11.5)	
	III	44	(24.6)	22	(29.3)	22	(21.2)	
	IV	3	(1.7)	3	(4.0)	--	(--)	
	IVA	38	(21.2)	6	(8.0)	32	(30.8)	
	IVB	26	(14.5)	7	(9.3)	19	(18.3)	
	IVC	1	(0.6)	--	(--)	1	(1.0)	

^‡^ *p* value from Pearson’s Chi square (χ^2^) tests; ECOG, Eastern Cooperative Oncology Group performance score; Charlson comorbidity score; LHSCC, laryngeal- and hypopharyngeal squamous cell carcinoma; OPSCC, oropharyngeal squamous cell carcinoma; OSCC, oral squamous cell carcinoma; T, Tumor size; N, Nodal involvement; M, distant metastasis; UICC, Union for International Cancer Control.

## Data Availability

The datasets presented in this article are not readily available because of patient confidentiality and participant privacy terms. Requests to access the datasets should be directed to G.W., Gunnar.Wichmann@medizin.uni-leipzig.de.

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
