# Peer review of "Pre-Therapeutic VEGF Level in Plasma Is a Prognostic Bio-Marker in Head and Neck Squamous Cell Carcinoma (HNSCC)"

_cancers, 2021, doi:10.3390/cancers13153781_

Round 1

Reviewer 1 Report

The manuscript measured vascular endothelial growth factor (VEGF) levels in serum and plasma samples of patients with head and neck cancer.  VEGF concentrations were significantly lower in plasma compared to serum.  Low plasma VEGF concentrations correlated with better clinical outcomes. 

In Table 2, the patient age categories are somewhat confusing.  As designated in the manuscript, one would expect increasing patient numbers with age.  Presumably the authors intended an age range such as <50 years, 50-59 years, 60-69 years, >70 years.

There may be a typographical error in line 193.  “Worser” is not standard English language.

There appears to be an error in document processing on p. 14 resulting in several lines of superimposed text.

Author Response

Dear reviewer, please find attached the PDF file containing the response to your comments highlighted blue.

Many thanks for your time reviewing the manuscript and contributing to now improved quality of the paper!

Gunnar Wichmann

Reviewer 2 Report

In the manuscript "Pre-therapeutic VEGF level in plasma is a prognostic bio-marker in head and neck squamous cell carcinoma (HNSCC)", Julia Siemert et al. analyzed VEGF levels in both plasma and serum of two independent cohorts of HNSCC patients. The work is well written and the aim of the study is of high interest for the oncology prospective. However, before the publication in Cancers, I have some comments that should be addressed.

The lack of OS significative difference in the TC cohort between patients stratified for the  VEGFplasma (<26ng/L and >26ng/L) limits the study novelty. I suggest modifying the title of the paper by adding that VEGF level in plasma is a “EFS biomarker” in HNSCC.  In this line, since the OS p value of the VEGF serum is lower than the plasma one, I would appreciate to see the Kaplan-Meier plots of the Figure 2 also for HNSCC patients stratified according to VEGF serum. Furthermore, to help the comprehension of the text, authors should add representative graphs and plots of data mentioned from lines 188 to lines 192.

Notably, a table displaying the Youden Indexes is missing. It will help to understand how a cut-off exceeding the CI (VEGFserum 264 ng/L) can be considered as predictive biomarker for HNSCC outcome. Additionally, the VEGF serum level of 24 patients (32%) exceed the CI; authors should comment it.

Author Response

(The authors gave the same response as above.)

Reviewer 3 Report

This paper presents data to answer a simple question: If we believe that VEGF is an important driver of angiogenesis that directly affects cancer progression, which is well supported in many scientists, then can the clinical usefulness of anti-angiogenic and possibly vascular disrupting agents for hard-to-treat cancers, such as HNSCC, be improved by better patient selection criteria? The authors attempt to answer this by putting all comers in a study to evaluate pre-treatment VEGF in serum and plasma samples to determine if circulating levels could predict a survival benefit for administration of VEGF inhibitors that would outweigh adverse events. They found a much lower level in plasma than serum, but plasma levels seemed to be a better predictor of EFS, NC and LRC, but not OS in the test cohort, compared to the independent validation cohort where there was a survival prediction, a finding that was maintained in the combined cohort.

So, the goal of this paper, even though very simplistic, looking at plasma vs serum and determining a cut off value that might be predictive, was achieved, there is a lot of patient data presented that might have yielded more detailed information, such as if the outcomes were at all influenced by treatment. Were patients given anti-angiogenics? Was there a difference in patients treated with chemoradiation or other modalities? Were any treated by proton beam instead of xrays? Could this data be used to predict outcomes to plan treatment regimens even if bevacizumab or other agents are not included?

Author Response

(The authors gave the same response as above.)

Reviewer 4 Report

I read with the interest manuscript entitled “Pre-therapeutic VEGF level in plasma is a prognostic bio-marker in head and neck squamous cell carcinoma (HNSCC).” The authors have found the differences between the level of VEGF in plasma and serum in therapy-naive patients with HNSCC. Moreover, they try to emphasize the role of VEGF in plasma as an independent predictor for outcome in HNSCC patients, which can be used in therapy planning in the future. However, I have to underline some major disadvantages of the paper:

  1. Abstract – information about study group is written only partially.
  2. Materials and methods – lack of explanation why these 4 particular clinical trial groups were included in the study.
  3. Tables 1 and 2 contain partially duplicate data. Please remove them or perform one common table.
  4. All the abbreviations in the tables have to be explained (f.ex. OP, RT).
  5. Significant differences between groups (p< 0.05) should be highlighted bold in all parts of the table.
  6. I was not able to read the text on the top of page nr 14 – it was overlapped.
  7. Manuscript contains few interpunctuation errors.

Thank you very much for your effort to perform the study and prepare the manuscript

Author Response

(The authors gave the same response as above.)

Round 2

Reviewer 2 Report

The paper is now suitable for the publication.